# Fine-Tuning Out-of-Vocabulary Item Recommendation with User Sequence Imagination

**Ruochen Liu[1], Hao Chen[2], Yuanchen Bei[3], Qijie Shen[4],**
**Fangwei Zhong[5], Senzhang Wang[1]\*, Jianxin Wang[1]**
[1]Central South University, [2]City University of Macau,
[3]Zhejiang University, [4]Alibaba Group, [5]Beijing Normal University
`{ruochen, szwang`\*`, jxwang}@csu.edu.cn, sundaychenhao@gmail.com`
`yuanchenbei@zju.edu.cn, qjshenxdu@gmail.com, fangweizhong@bnu.edu.cn`

## Abstract

Recommending out-of-vocabulary (OOV) items is a challenging problem since the in-vocabulary (IV) items have well-trained behavioral embeddings but the OOV items only have content features. Current OOV recommendation models often generate 'makeshift' embeddings for OOV items from content features and then jointly recommend with the 'makeshift' OOV item embeddings and the behavioral IV item embeddings. However, merely using the 'makeshift' embedding will result in suboptimal recommendation performance due to the substantial gap between the content feature and the behavioral embeddings. To bridge the gap, we propose a novel **U**ser **S**equence **IM**agination (**USIM**) fine-tuning framework, which first imagines the user sequences and then refines the generated OOV embeddings with the user behavioral embeddings. Specifically, we frame the user sequence imagination as a reinforcement learning problem and develop a recommendation-focused reward function to evaluate to what extent a user can help recommend the OOV items. Besides, we propose an embedding-driven transition function to model the embedding transition after imaging a user. USIM has been deployed on a prominent e-commerce platform for months, offering recommendations for millions of OOV items and billions of users. Extensive experiments demonstrate that USIM outperforms traditional generative models in OOV item recommendation performance across traditional collaborative filtering and GNN-based collaborative filtering models.

## 1 Introduction

Recommendation systems, such as collaborative filtering models, learn behavioral embeddings from historical interactions to represent the behavioral characteristics of billions of users and items [1–4]. For instance, the embeddings of interacted user-item pairs have higher inner products, whereas those of un-interacted pairs have lower inner products. However, besides the items with user interactions, thousands of out-of-vocabulary (OOV) items—such as short videos, photos, and posts—are generated or uploaded every second. In the age of AGI, the generation speed of AI-made OOV content, including text, images, and videos, will far exceed the speed of human consumption. To avoid being overwhelmed by the OOV content, it is essential to replicate how humans handle these items, recommending them without disrupting in-vocabulary (IV) items.

Traditional OOV recommendation models usually generate 'makeshift' embeddings from the content features and then use them to recommend OOV items. The research can be classified into two categories. (a) **Generative models** aim to generate realistic embeddings. GAR [5] uses a generative

---

*Corresponding author

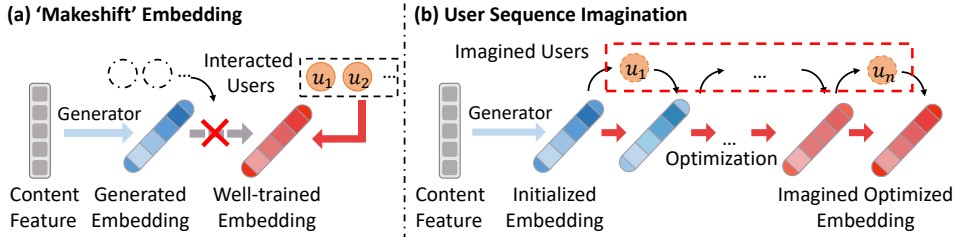

Figure 1: Comparison between (a) traditional 'makeshift' embedding OOV recommendation framework, and (b) user sequence imagination OOV recommendation framework.

adversarial structure to ensure the embedding distribution of generated OOV embeddings is similar to IV embeddings. ALDI [6] distills knowledge from IV items to OOV items. (b) **Dropout models** increase the robustness of recommender systems. Dropout [7] randomly substitutes IV embeddings with "makeshift" ones to enhance system robustness. Heater [8] and CLCRec [9] further utilize a mix of experts and contrastive learning techniques to improve OOV recommendation performance.

As shown on the left of Figure 1, current recommender systems typically learn the embeddings for any given items by initializing/generating the embedding and then optimizing them through user-sequence backpropagation. However, prevalent OOV recommendation models primarily concentrate on generating improved or robust embeddings, overlooking the potential for further optimization from imagining user sequences. This oversight limits these models due to the following issues. 1. **Content-Behavior Gap**. 'Makeshift' embeddings are generated from content features, while behavioral embeddings are trained using backpropagation. The substantial difference between content features and behavioral embeddings may lead to discrepancies between IV and OOV items, impacting IV, OOV, or both. 2. **Potential Suboptimality**. Focusing only on embedding generation overlooks potential improvements from backpropagation, which could finely tune the embeddings to adapt to user preferences and current recommender systems, possibly leading to suboptimal recommendation performance and a reduction in revenue.

While imagining the sequential optimization process for OOV item embeddings shows promise, its implementation presents three challenges. 1. **Absence of Historical Interactions.** The lack of historical user interactions for OOV item embeddings hinders the definition of a clear backpropagation and optimization process. 2. **Ambiguous Imagination Objectives.** Formulating objectives, stopping criteria, and user selection for imagining OOV item interactions remains an open challenge. 3. **Navigating the Vast User Space.** Efficiently identifying suitable user sequences to imagine for OOV items within the massive user space of recommender systems.

To address these challenges, we introduce an **User Sequence Imagination (USIM)** pipeline that further optimizes the embedding of OOV items. It first imagines potential users who may interact with the OOV item and then refines the item embeddings through backpropagation. Specifically, we propose a **RL-based USIM** solution, which formulates the sequential optimization as a Markov Decision Process and introduces recommender-oriented PPO (RecPPO) to maximize the final recommendation performance of OOV items. In summary, our contributions are as follows:

- We introduce USIM, a novel approach that fundamentally addresses the Out-of-Vocabulary (OOV) problem by imagining user sequences and performing user-sequence backpropagation.

- We formally define the Reinforcement Learning (RL) formulation of USIM, including the formulation of the Markov Decision Process (MDP) for user sequence imagination, and provide the state transition function framework for user-sequence backpropagation.

- We implement USIM on a major e-commerce platform—Alibaba, successfully optimizing millions of OOV items and recommending them to billions of users. The source code is publicly available at `https://github.com/Ruochen1003/USIM`.

- We validate the effectiveness of our approach on two benchmark datasets using both traditional collaborative filtering and graph-based collaborative filtering backbones. Extensive experiments demonstrate that USIM outperforms existing state-of-the-art OOV recommendation models in terms of OOV recommendation performance and overall recommendation quality.

## 2 Preliminaries

**Notations.** Let $\mathcal{U}$ and $\mathcal{I}$ denote the sets of users and items, respectively. We partition $\mathcal{I}$ into in-vocabulary (IV) items $\mathcal{I}_{iv}$ (items with interaction history) and out-of-vocabulary (OOV) items $\mathcal{I}_{oov}$ (items without interaction history). We denote the cardinality of these sets as $|\mathcal{U}|$, $|\mathcal{I}_{iv}|$, and $|\mathcal{I}_{oov}|$, respectively. For clarity in the following discussion, we also use $\mathcal{U}_i$ to represent the set of users who have interacted with item $i$.

The embedding matrices for users, IV items, and OOV items are denoted as $\boldsymbol{E}_u \in \mathbb{R}^{|\mathcal{U}| \times d}$, $\boldsymbol{E}_{iv} \in \mathbb{R}^{|\mathcal{I}_{iv}| \times d}$, and $\boldsymbol{E}_{oov} \in \mathbb{R}^{|\mathcal{I}_{oov}| \times d}$, respectively, where $d$ represents the embedding dimension. For an individual user $u$ and item $i$, their corresponding embeddings are denoted as $\boldsymbol{e}_u \in \mathbb{R}^d$ and $\boldsymbol{e}_i \in \mathbb{R}^d$. Since OOV items lack behavioral embeddings initially, we leverage content features, denoting the content feature vector of item $i$ as $\boldsymbol{c}_i$.

**Backpropagated Embedding of IV Items.** The embeddings of IV items are initialized using standard initialization techniques such as Xavier initialization [10]. These embeddings are subsequently optimized through backpropagation using historical interaction data [11, 4],

$$\boldsymbol{e}_i = \boldsymbol{e}_i - \sum_{u \in \mathcal{U}_i} \nabla L(u, i), \tag{1}$$

where $L$ represents the loss function (e.g., BPR loss).

**Makeshift Embedding of OOV Items.** Due to the absence of historical interactions for OOV items, various approaches have been proposed to generate makeshift embeddings that enable joint recommendation with IV items. These approaches can be broadly categorized into generative models [6, 5, 12] and dropout models [9, 7], which transform content features $\boldsymbol{c}_i$ into embeddings through a generator function $G$,

$$\boldsymbol{e}_i = G(\boldsymbol{c}_i), \tag{2}$$

where $G$ is optimized using various objective functions, including similarity-based losses [12], adversarial losses [5], and knowledge distillation losses [6].

**MDP Formulation of Back Propagation.** We formulate the OOV embedding optimization process as a Markov Decision Process (MDP) to narrow the optimization gap between IV and OOV item embeddings. This formulation enables simultaneous user imagination and embedding optimization through backpropagation.

An MDP at time step $t$ is defined by the quintuple $(\mathcal{S}, \mathcal{A}, \rho, R, \gamma)$, where:

- $\mathcal{S}$ represents the state space, with each state $\boldsymbol{s} \in \mathcal{S}$ capturing the environment configuration
- $\mathcal{A}$ denotes the action space, where each action $a \in \mathcal{A}$ represents a possible agent decision
- $\rho : \mathcal{S} \times \mathcal{A} \to \mathcal{S}$ defines the state transition function, with $\boldsymbol{s}_{t+1} = \rho(\boldsymbol{s}_t, a_t)$
- $R : \mathcal{S} \times \mathcal{A} \to \mathbb{R}$ specifies the reward function, where $r_t = R(\boldsymbol{s}_t, a_t)$
- $\gamma \in [0, 1]$ represents the discount factor for future rewards

## 3 Proposed User Sequence Imagination Model

### 3.1 Framework Overview

To bridge the fundamental gap between IV and OOV item embedding generation caused by disparate interaction histories, we propose USIM, which fine-tunes OOV item embedding by imagining appropriate user sequences. Specifically, in each step, USIM simulates a user interaction and optimizes the item embedding accordingly. We formulate this process within a reinforcement learning paradigm, as illustrated in Figure 2.

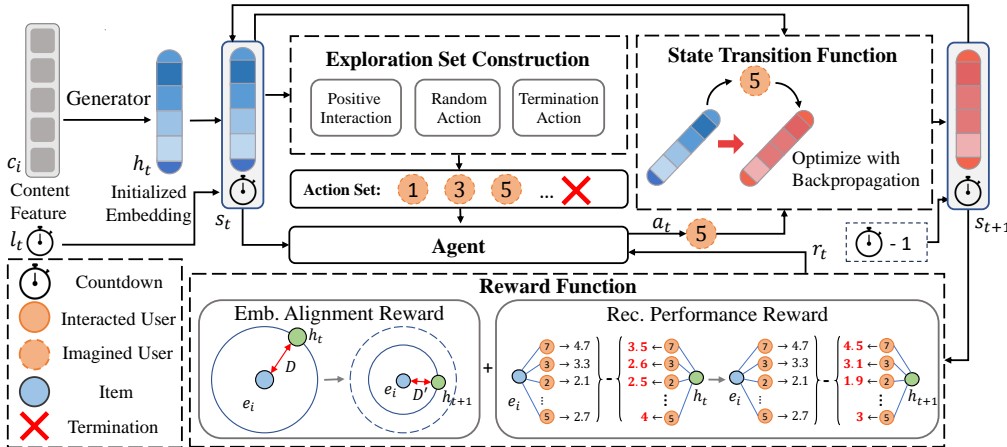

Figure 2: The overview framework of USIM. USIM fine-tunes the generated OOV item embeddings through sequential user interaction imagination, guided by exploration set construction, state transition, and a tailored reward mechanism.

**State Space.** At time step $t$, the state $s_t$ encapsulates the essential information required for item embedding optimization, defined as $s_t = [h_t, l_t]$. The state representation $h_t \in \mathbb{R}^d$, which is refined through optimization, resides in the same embedding space as $e_i$, and its final representation will serve as the OOV item embedding. To specifically denote the state of item $i$ at time step $t$, we define $s_{i,t} = [h_{i,t}, l_{i,t}]$. In contrast, $s_t$ serves as a more general denotation. The initial state presentation is obtained by the generator in Eq. (2) as $h_{i,0} = G(c_i)$. Details can refer to Appendix B.

The temporal component $l_t$ is used as a countdown mechanism, tracking the remaining optimization steps to encourage efficient convergence [13]. Given a maximum action limit $N$, at time step $t$, the countdown value is computed as $l_t = N - t$.

**Action Space.** Given state $s_t$, the agent selects an action $a_t$ from the action space $\mathcal{A} = \mathcal{U} \cup \{a_{end}\}$, where $a_{end}$ denotes the termination action. This selection process entails either imagining a user or terminating the optimization process. The action embedding $e_a$ corresponds to the user embedding from $E_u$ when $a \in \mathcal{U}$, and defaults to $\mathbf{0}$ for the termination action.

**Policy Network.** The agent's decision-making process is governed by policy $\pi(s_t)$, which maps the current state $s_t$ to a probability distribution over possible actions. To effectively utilize existing embeddings while accommodating the special termination action, the policy distribution is given as,

$$\pi(a_t|s_t) = \begin{cases} (1 - \sigma(\boldsymbol{W}_2 \boldsymbol{s}_t^\top + c)) \cdot \frac{\exp(\boldsymbol{e}_{a_t} \boldsymbol{W}_1 \boldsymbol{s}_t^\top)}{\sum_{a \in \mathcal{U}} \exp(\boldsymbol{e}_a \boldsymbol{W}_1 \boldsymbol{s}_t^\top)}, & \text{if } a_t \in \mathcal{U}; \\ \sigma(\boldsymbol{W}_2 \boldsymbol{s}_t^\top + c), & \text{if } a_t = a_{end}, \end{cases} \tag{3}$$

where $\boldsymbol{W_1} \in \mathbb{R}^{d \times (d+1)}$, $\boldsymbol{W_2} \in \mathbb{R}^{d+1}$, and $c$ are parameters, and $\sigma$ represents the sigmoid function.

**Reward and State Transition.** The agent receives an immediate reward $r_t = R(s_t, a_t)$ after each action, guiding the optimization trajectory. To align with the imagination process, we design an efficient state transition function $s_{t+1} = \rho(s_t, a_t)$ that facilitates the progressive refinement from content-based to interaction-based embeddings. The experience tuples $(s_t, a_t, r_t, s_{t+1})$ are collected in a replay buffer for subsequent training iterations.

### 3.2 State Transition Function

State transition involves modifying $h_t$, which will ultimately be used as the OOV item embedding and each step of the state transition corresponds to optimizing $h_t$ using the imagined user(last action $a_t$). Therefore, to design a state transition function that aligns with this optimization process, we must

first determine the objective of this optimization. Considering that most current recommendation algorithms calculate relevance scores between users and items for recommendations [11, 14, 4], we adopt the following objective as our optimization goal,

$$\min_{\boldsymbol{e}_i} -\mathbb{E}_{u \in \mathcal{U}_i} \hat{y}_{u,i}, \tag{4}$$

where $\hat{y}_{u,i}$ is predicted score between $\boldsymbol{e}_i$ and $\boldsymbol{e}_u$.

Based on the above idea, we can view the user imagination process as the solution to the optimization objective (Eq. (4)). Specifically, we assume that the users imagined by the agent are the users who have interacted with this item, and by using backpropagation, we can optimize the content-based initialized embeddings. Thus, the transition function $\rho_h$ of $\boldsymbol{h}_t$ can be written as follows,

$$\boldsymbol{h}_{t+1} = \rho_h(\boldsymbol{h}_t, a_t) = \begin{cases} \boldsymbol{h}_t + \lambda \nabla \hat{y}_{a_t,i} & \text{if } a_t \in \mathcal{U}; \\ \boldsymbol{h}_t & \text{if } a_t = a_{end}, \end{cases} \tag{5}$$

where $\lambda$ is a hyperparameter and can be understood as the learning rate. If $\hat{y}_{a_t,i}$ is computed using dot product, the final transition function of $\boldsymbol{h}_t$ can be written as follows,

$$\boldsymbol{h}_{t+1} = \rho_h(\boldsymbol{h}_t, a_t) = \begin{cases} \boldsymbol{h}_t + \lambda \cdot \boldsymbol{e}_{a_t} & \text{if } a_t \in \mathcal{U}; \\ \boldsymbol{h}_t & \text{if } a_t = a_{end}. \end{cases} \tag{6}$$

And $l_t$ can be updated as $l_{t+1} = l_t - 1$.

## 3.3   Reward Function

The objective of the USIM is to optimize the initialized embeddings by imagining user sequences. To facilitate this process, we design a reward function to guide the reinforcement learning approach. Our reward function consists of three components.

**Embedding Alignment Reward**. We believe that the item embeddings generated by the IV model represent the best solution for our optimization process. Consequently, our objective is to closely align the final state representation with actual item embedding (i.e., the corresponding embedding from the IV model). To achieve this, we calculate the reward based on the concept of similarity,

$$R_{emb}(\boldsymbol{h}_{i,t}, a_t) = D(\boldsymbol{h}_{i,t}, \boldsymbol{e}_i) - D(\boldsymbol{h}_{i,t+1}, \boldsymbol{e}_i), \boldsymbol{h}_{i,t+1} = \rho_h(\boldsymbol{h}_{i,t}, a_t), \tag{7}$$

where $D(\cdot, \cdot)$ denotes the Euclidean distance between embeddings. This reward represents the change in similarity between the state representation and the actual item embedding, before and after the state transition.

**Recommendaion Performance Reward**. Although the embedding alignment reward encourages the state representation $\boldsymbol{h}_t$ to be close to the actual item embedding, it does not differentiate between states when multiple representations are equally distant from the actual embedding. Additionally, it does not fully utilize the insights from existing user interactions. Therefore, we design a reward function based on recommendation performance as follows,

$$R_{rec}(\boldsymbol{h}_{i,t}, a_t) = f(\boldsymbol{h}_{i,t}, \boldsymbol{e}_i) - f(\boldsymbol{h}_{i,t+1}, \boldsymbol{e}_i), \boldsymbol{h}_{i,t+1} = \rho_h(\boldsymbol{h}_{i,t}, a_t),$$
$$f(\boldsymbol{h}_{i,t}, \boldsymbol{e}_i) = \frac{1}{|\mathcal{U}_i|} \sum_{u_j \in \mathcal{U}_i} |\boldsymbol{h}_{i,t} \cdot \boldsymbol{e}_{u_j} - \boldsymbol{e}_i \cdot \boldsymbol{e}_{u_j}|, \tag{8}$$

where $f(\boldsymbol{h}_{i,t}, \boldsymbol{e}_i)$ represents the predictive power of state representation $\boldsymbol{h}_{i,t}$ for users in $\mathcal{U}_i$. The final performance reward is derived from the change in $f$ before and after the state transition. This change represents the variation in the embedding's predictive capability.

**Step Regulation**. To encourage the agent to achieve the goal in as few steps as possible, we impose a penalty for each action it takes. Therefore, the final reward function is as follows,

$$r_t = R(\boldsymbol{s}_{i,t}, a_t) = R_{emb}(\boldsymbol{h}_{i,t}, a_t) + R_{rec}(\boldsymbol{h}_{i,t}, a_t) - p, \tag{9}$$

where $p$ is a hyperparameter that represents the penalty.

## 3.4 Exploration Set Construction

Given the large user base in the recommendation dataset, sampling actions solely by probability during initial reinforcement learning often results in negative rewards, slowing convergence and reducing performance. To explore actions more efficiently, we construct an exploration set according to state $s_t$, comprising three components.

**Positive Action**. In the USIM framework (Section 3.2), the agent assumes that the imagined users correspond to those who have interacted with the item, so these users are included in the exploration set. While optimizing $h_{i,t}$ to match $e_i$ theoretically requires only the interaction set $\mathcal{U}_i$, achieving this often demands combining multiple actions $a \in \mathcal{U}_i$, making exploration complex. To streamline this, we select users most likely to bridge $h_{i,t}$ and $e_i$, accelerating the process. The Positive Action set $\mathcal{U}_{pos}$ is constructed as follows,

$$\mathcal{U}_{pos} = \text{Top}_{k_1}((e_i - h_{i,t}), e_u) \cup \mathcal{U}_i, \tag{10}$$

where $\text{Top}_{k_1}((e_i - h_{i,t}), e_u)$ represents the set of $k_1$ users with the highest cosine similarity to $e_i - h_{i,t}$, forming a vector that directly points to the actual item embedding.

**Random Action**. Relying solely on the aforementioned action sets may overly restrict actions, limiting state space coverage and reducing model generalization. Moreover, sampling actions with negative rewards can also benefit training [15]. Thus, we augment the action set by randomly selecting $k_2$ actions from the remaining pool, denoted as $\mathcal{U}_{rad}$.

**Termination Action**. To enable the agent to learn when to terminate, we also incorporate the termination action $a_{end}$ into the final action set. This inclusion allows the agent to determine the appropriate timing for ending the optimization process.

For simplicity, we set $k_1 = k_2 = k$. And exploration set $\mathcal{A}_{samp}$ can be represented as follows,

$$\mathcal{A}_{samp} = \mathcal{U}_{pos} \cup \mathcal{U}_{rad} \cup \{a_{end}\}. \tag{11}$$

During the training phase, as the sampling range is narrowed from the full action set $\mathcal{A}$ to a specific action set $\mathcal{A}_{samp}$, we rewrite the Eq. (3) as follows,

$$\pi(a_t|s_t) = \begin{cases} (1 - \sigma(\boldsymbol{W_2}\boldsymbol{s}_t^\top + c)) \cdot \frac{\exp\{\boldsymbol{e}_{a_t}\boldsymbol{W_1}\boldsymbol{s}_t^\top\}}{\sum_{a \in A_{samp} \setminus \{a_{end}\}} \exp\{\boldsymbol{e}_a\boldsymbol{W_1}\boldsymbol{s}_t^\top\}}, & \text{if } a_t \in A_{samp} \setminus \{a_{end}\}; \\ \sigma(\boldsymbol{W_2}\boldsymbol{s}_t^\top + c), & \text{if } a_t = a_{end}; \\ 0, & \text{if } a_t \notin \mathcal{A}_{samp}. \end{cases} \tag{12}$$

## 3.5 Training with RecPPO

We incorporate recommendation-specific supervision signals into PPO [16], referring to this enhanced approach as Recommender-Oriented PPO (RecPPO), to train our USIM. In the scenario of OOV item recommendation, the optimal action following certain states is clear, allowing the cumulative expected rewards for these states to be calculated directly. When the state representation $h_{i,t}$ of a specific item is equal to its item embedding $e_i$, according to our designed reward function, the expected value should be 0 because any subsequent actions, except for termination will lead to negative rewards.

So we use these supervision signals to assist in training the value network $V_\omega$, the specific loss function of the value network is defined as follows,

$$\mathcal{L}(\omega) = \frac{1}{|B|} \sum_{(s_t, r_t, s_{t+1}) \in B} \left[ (r_t + \gamma V_\omega(s_{t+1}) - V_\omega(s_t))^2 \right] + \frac{1}{|\mathcal{I}|} \sum_{i \in \mathcal{I}} V_\omega([e_i, \text{random}(0, N)])^2, \tag{13}$$

where $B$ denotes tuples sampled from the buffer pool, and $\text{random}(0, N)$ is a random number between 0 and $N$. The first term of the loss is the Temporal Difference loss used in value network training, while the second term includes our recommendation-oriented supervision signals. Regardless of previous actions, when the state representation $h_t^i$ matches $e_i$, the agent should terminate. Here, $\text{random}(0, N)$ acts as the timer for each termination state. As for the policy network, we train it using the same method as PPO. The whole training process can be found in Appendix C.

# 4 Experiments

We conduct comprehensive experiments on two benchmark datasets aiming to address the following three questions: **RQ1:** Can USIM achieve superior OOV item recommendation performance compared to state-of-the-art OOV item recommendation models? **RQ2:** How key components of USIM affect its performance? **RQ3:** Is the proposed USIM more effective than representative RL methods? **RQ4:** What is the tendency of performance during USIM's imagination process? **RQ5:** How does USIM perform in real-world industrial recommendations? **RQ6:** How does USIM achieve efficiency compared to other baselines?

## 4.1 Experimental Setup

Table 1: Overall, OOV, and IV item recommendation performance comparison over two representative recommender backbones (MF and GNN). The best and second-best results in each column are highlighted in **bold** font and underlined, $\star\star$ indicates the statistical significance $p < 0.01$ compared to the best-performed baseline, $\star$ indicates the statistical significance $p < 0.05$ compared to the best-performed baseline.

| | Method | Overall Recommendation | | | | OOV Recommendation | | | | IV Recommendation | | | |
|---|---|---|---|---|---|---|---|---|---|---|---|---|---|
| | | CiteULike | | MovieLens | | CiteULike | | MovieLens | | CiteULike | | MovieLens | |
| | | Recall | NDCG | Recall | NDCG | Recall | NDCG | Recall | NDCG | Recall | NDCG | Recall | NDCG |
| MF | DropoutNet | 0.0973 | 0.0768 | 0.1148 | 0.1898 | 0.2442 | 0.1485 | 0.1352 | 0.1417 | 0.1560 | 0.0941 | 0.2577 | 0.2442 |
| | MTPR | 0.1090 | 0.0822 | 0.1210 | 0.1936 | 0.2489 | 0.1427 | 0.1364 | 0.1408 | 0.1808 | 0.1021 | 0.2463 | 0.2233 |
| | Heater | 0.1149 | 0.0881 | 0.1163 | 0.1901 | 0.2578 | 0.1532 | 0.0601 | 0.0734 | 0.1674 | 0.0999 | 0.2604 | 0.2640 |
| | CLCRec | 0.1242 | 0.1047 | 0.1101 | 0.1556 | 0.2187 | 0.1331 | 0.0774 | 0.0751 | 0.2197 | 0.1411 | 0.2413 | 0.2089 |
| | DeepMusic | 0.1451 | 0.1287 | 0.1359 | 0.2153 | 0.2194 | 0.1281 | 0.0978 | 0.1003 | **0.2838** | **0.1933** | **0.3023** | **0.2811** |
| | MetaEmb | 0.1326 | 0.1210 | 0.1359 | 0.2153 | 0.2203 | 0.1307 | **0.1417** | 0.1445 | **0.2838** | **0.1933** | **0.3023** | **0.2811** |
| | GAR | 0.1440 | 0.1132 | 0.0462 | 0.0812 | 0.2453 | 0.1479 | 0.0348 | 0.0510 | 0.2272 | 0.1438 | 0.1003 | 0.1003 |
| | ALDI | 0.1584 | 0.1212 | 0.1360 | 0.2154 | 0.2622 | 0.1542 | 0.1258 | 0.1262 | **0.2838** | **0.1933** | **0.3023** | **0.2811** |
| | **USIM** | **0.1926**\*\* | **0.1530**\*\* | **0.1369**\* | **0.2164**\* | **0.2753**\*\* | **0.1647**\*\* | 0.1401 | **0.1466**\*\* | **0.2838** | **0.1933** | **0.3023** | **0.2811** |
| | %Improv. | 19.00% | 10.96% | 0.66% | 0.46% | 5.00% | 6.81% | - | 1.45% | - | - | - | - |
| GNN | DropoutNet | 0.0914 | 0.0718 | 0.1205 | 0.1962 | 0.2337 | 0.136 | 0.1552 | 0.1506 | 0.1596 | 0.0949 | 0.2713 | 0.2530 |
| | MTPR | 0.1066 | 0.0783 | 0.1217 | 0.1949 | 0.2309 | 0.1355 | 0.1412 | 0.1373 | 0.1749 | 0.0963 | 0.2708 | 0.2520 |
| | Heater | 0.1116 | 0.0869 | 0.1236 | 0.2053 | 0.2467 | 0.1439 | 0.0366 | 0.0319 | 0.1703 | 0.1007 | 0.2746 | 0.2647 |
| | CLCRec | 0.1303 | 0.1150 | 0.1127 | 0.1656 | 0.2315 | 0.1293 | 0.0535 | 0.0627 | 0.2260 | 0.1537 | 0.2515 | 0.2226 |
| | DeepMusic | 0.1244 | 0.1058 | 0.1428 | 0.2353 | 0.1899 | 0.1131 | 0.1539 | 0.148 | **0.2685** | **0.1723** | **0.3183** | **0.3048** |
| | MetaEmb | 0.1249 | 0.1088 | 0.1428 | 0.2353 | 0.2150 | 0.1249 | 0.1560 | 0.1567 | **0.2685** | **0.1723** | **0.3183** | **0.3048** |
| | GAR | 0.1098 | 0.0863 | 0.0179 | 0.0311 | 0.2053 | 0.1024 | 0.0354 | 0.0451 | 0.2364 | 0.1485 | 0.0288 | 0.0268 |
| | ALDI | 0.1371 | 0.1013 | 0.1429 | 0.2354 | 0.2466 | 0.1399 | 0.1358 | 0.144 | **0.2685** | **0.1723** | **0.3183** | **0.3048** |
| | **USIM** | **0.1629**\*\* | **0.1199**\*\* | **0.1448**\*\* | **0.2377**\*\* | **0.2493**\*\* | **0.1452**\*\* | **0.1564** | **0.1638**\*\* | **0.2685** | **0.1723** | **0.3183** | **0.3048** |
| | %Improv. | 5.90% | 4.26% | 1.32% | 0.98% | 0.93% | 1.65% | 0.26% | 4.53% | - | - | - | - |

**Datesets**. We evaluate the performance of USIM on OOV items using the widely used datasets: CiteULike [8] and MovieLens [17]. Specifically, CiteULike contains 5,551 users, 16,980 articles (items), and 204,986 interactions. MovieLens comprises 6,040 users, 3,883 movies (items), and 1,000,210 interactions. Details about these datasets are shown in Appendix D.1.

**Baselines**.To assess the effectiveness and universality of USIM, we conduct a comparative analysis with 8 leading-edge models in OOV item recommendations across two distinct datasets. These models include two main groups. (i) Dropout-based methods: DropoutNet [7], MTPR [18], Heater [8], and CLCRec [9]. (ii) Generative-based methods: DeepMusic [12], MetaEmb [19], GAR [5], and ALDI [6]. Details about these models are shown in Appendix D.2. To further verify the generalization ability, we adopted both the widely used collaborative filtering model MF [20] and GNN-based model NGCF [3] as the recommender, respectively.

**Evaluation Metrics**. Following the evaluation of existing OOV recommendation [6, 9], we conduct three different tasks in our experiments: (1) Overall Recommendation, (2) OOV Recommendation, and (3) IV Recommendation. We employ the full-ranking evaluation approach to assess the performance of overall, IV, and OOV recommendations. Following previous works [4, 3], we utilize Recall@K and Normalized Discounted Cumulative Gain (NDCG@K) as metrics.

**Implementation Details**. We implement the baselines using their officially provided version. See Appendix E for the detailed implementation. The best hyperparameters are found for each dataset. For fairness, we use the same options and follow the designs in their articles for all baselines.

Table 2: Ablation study results between USIM with its four variants on CiteULike.

| Variant | MF | | | | GNN | | | |
| | Overall | | OOV | | Overall | | OOV | |
| | Recall | NDCG | Recall | NDCG | Recall | NDCG | Recall | NDCG |
|---|---|---|---|---|---|---|---|---|
| *w/o ct* | 0.1742 | 0.1439 | 0.2336 | 0.1411 | 0.1590 | 0.1205 | 0.2363 | 0.1198 |
| *w/o ra* | 0.1754 | 0.1433 | 0.2341 | 0.1412 | 0.1368 | 0.1156 | 0.2411 | 0.1336 |
| *w/o es* | 0.1436 | 0.1279 | 0.2232 | 0.1316 | 0.1221 | 0.1000 | 0.1971 | 0.1103 |
| *w/o pr* | 0.1866 | 0.1510 | 0.2711 | 0.1596 | 0.1417 | 0.1175 | 0.2414 | 0.1344 |
| **USIM** | **0.1926** | **0.1530** | **0.2753** | **0.1647** | **0.1632** | **0.1245** | **0.2534** | **0.1478** |

## 4.2 Main Results (RQ1)

The main comparison results of overall, IV, and OOV item recommendation results can be found in Table 1. From the results, we can have the following observations.

**USIM can generally achieve significant improvements over state-of-the-art methods on both overall and OOV item recommendations while keeping the IV item recommendation.** From the tables, we observe that the USIM achieves the highest average Recall and NDCG performance across both MF and GNN recommenders. These comparison results verify the superiority of the imagined embeddings over traditional one-step generated embeddings.

**Dropout-based baselines have the performance drop in the IV item recommendation.** We find that the USIM and the generative-based models can keep the IV item recommendation. However, dropout-based models will lead to a performance drop on IV items. This suggests that there is a difference between OOV items and IV items. Pre-training representations of IV items in advance to generate representations for OOV items may be better for retaining information for IV items.

## 4.3 Ablation Study (RQ2)

To validate the effectiveness of the individual components in our model, we compared the full model against four variants: (i) *w/o ct* removes the cosine similarity-based top-k user selection when constructing the positive action set. (ii) *w/o ra* removes the randomly sampled actions in the exploration set. (iii) *w/o es* does not construct an exploration set and directly explores the entire action set. (iv) *w/o pr* removes the performance reward, only using the similarity reward in the RL processing. From the results in Table 2, we make the following observations.

**Effectiveness of components in the exploration set construction**. The *w/o ct* result indicates that the cosine similarity-based selection of the top-$k$ users can simplify the exploration process by quickly identifying users related to the item embedding. Then, the performance of *w/o ra* reveals the importance of including negative samples to provide a more comprehensive exploration signal, preventing the agent from overestimating action values by only considering positive feedback. Further, the poor performance of the *w/o es* approach highlights the challenge of effectively exploring an extremely large action space, emphasizing the need for a well-designed exploration strategy to guide the agent towards promising actions.

**Effectiveness of components in the reward function**, Removing the performance reward component leads to inferior results. This suggests that the reward function should not only consider the distance between the generated embedding and the target embedding but also explicitly incorporate the downstream recommendation performance of the generated embedding.

## 4.4 Comparison with Representative RL Methods (RQ3)

In Figure 3, we compared our proposed model with our initialized MLP and two other traditional reinforcement learning methods: Wolpertinger Policy [21] (WP) and Hierarchical Reinforcement Learning [22](HRL). WP utilizes the similarity of action representations and value estimation to solve the large discrete action space problem in reinforcement learning, while HRL employs a hierarchical decomposition of action space and sub-task solution to improve exploration efficiency.

Our model significantly outperforms the other two methods in both OOV recommendation and overall recommendation. Moreover, it can be observed that the other two methods do not show

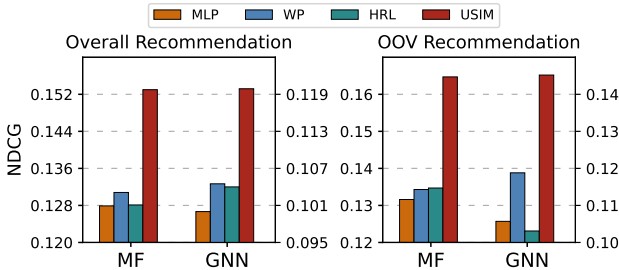

Figure 3: Comparing USIM with other RL methods for overall and OOV recommendation performance in CiteULike dataset.

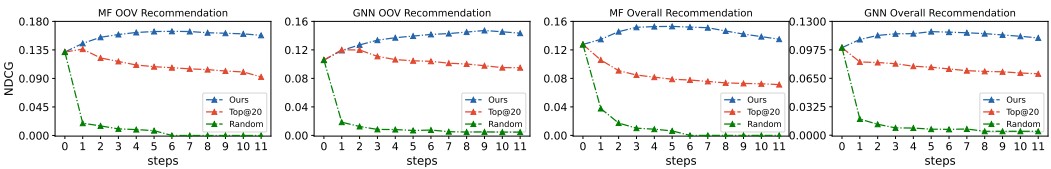

Figure 4: Performance analysis of different generation methods on the CiteUlike dataset.

apparent improvements compared to their initial states. This supports the effectiveness of our tailored exploration method in improving performance in OOV recommendations.

## 4.5 Case Study (RQ4)

To investigate the performance trend of our proposed method during the optimization process, and to validate the importance of the way of sampling in the embedding optimization process, we compare USIM with the following two user selection strategies: randomly sampling users at each step, and randomly selecting from the top 20 users with the highest relevance scores at each step. The results are shown in Figure 4, and according to the result, we can draw the following conclusions.

(i) Our method outperforms the random selection of top-20 users and the entire user set in both OOV and overall scenarios, indicating it can effectively identify users beneficial for optimization. However, performance declines as more users are imagined, likely due to reduced exploration by the agent.

(ii) When randomly selecting users from the top 20, we can observe that the performance in OOV recommendation increases after the first step. This suggests that sampling from high-scoring users is more likely to select users who are beneficial to the optimization process. However, the performance then continuously declines, indicating that this approach is not suitable for all states and has limited help for the optimization process.

(iii) Randomly selecting a user at each step leads to a drastic and non-recoverable decline in performance in both the OOV and overall scenarios after the first step. This suggests that in recommendation scenarios, it is extremely difficult to sample users from the massive user set who are beneficial to the optimization, and the majority of users are highly detrimental to the optimization process.

More experimental hyperparameter analysis can be found in Appendix F.

## 4.6 Online Evaluation (RQ5)

To evaluate the performance of USIM in an industrial setting, we conducted a two-week online A/B test on a major e-commerce platform with 5% of users in each group. USIM was compared against three baselines: Random, MetaEmb [19], and ALDI [6]. Details about our test platform and evaluation metrics are provided in Appendix G. Table 3 presents the results of these online A/B tests.

These remarkable improvements across all metrics underscore the effectiveness of the USIM in addressing the OOV item recommendation problem in real-world recommender systems. The

Table 3: Results of online A/B test in the industrial platform.

| A/B Test | OOV Item PV | OOV Item PCTR | OOV Item GMV |
|---|---|---|---|
| vs. Random | 8.20% | 2.80% | 20.30% |
| vs. MetaEmb | 6.55% | 1.95% | 14.95% |
| vs. ALDI | 4.90% | 1.10% | 13.60% |

consistent and substantial performance gains, particularly in OOV item GMV, highlight the practical impact of our approach on business outcomes in e-commerce settings.

### 4.7 Efficiency Analysis (RQ6)

To evaluate the time efficiency of USIM, especially in comparison to SOTA baselines, we recorded the total training time(Training Time), total convergence epochs(Converge Epochs), Time Per Epoch, and Inference Time for USIM and each baseline on the CiteULike and MovieLens datasets. The results are presented in Table 4. Based on these results, we can draw the following conclusions:

Table 4: Results of time efficiency.

| | Method | Training Time | Converge Epochs | Time Per Epoch | Inference Time |
|---|---|---|---|---|---|
| CiteULike | MetaEmb | 254s | 28 | 9.07s | 0.012s |
| | ALDI | 225s | 51 | 4.41s | 0.013s |
| | Heater | 841s | 57 | 14.75s | 0.074s |
| | CLCRec | 926s | 70 | 13.22s | 0.070s |
| | USIM | 484s | 36 | 13.44s | 0.031s |
| MovieLens | MetaEmb | 415s | 11 | 37.72s | 0.005s |
| | ALDI | 664s | 50 | 13.28s | 0.004s |
| | Heater | 1474s | 20 | 73.7s | 0.072s |
| | CLCRec | 3684s | 59 | 62.44s | 0.084s |
| | USIM | 330s | 45 | 7.33s | 0.045s |

(i) **USIM is Faster than Heater and CLCRec**: USIM computes embeddings only for OOV items, whereas Heater and CLCRec must compute embeddings for both OOV and IV items.

(ii) **USIM is Comparable with MetaEmb and ALDI**: USIM imagines sequences only for OOV items, resulting in inference times comparable to MetaEmb and ALDI.

(iii) **USIM is Efficient in Training**: By fundamentally addressing OOV recommendation, USIM converges in fewer epochs, making training more efficient.

More experiments about online recommendation efficiency can be found in Appendix H.

## 5  Conclusion

Recommending out-of-vocabulary (OOV) items is challenging due to the lack of well-trained behavioral embeddings. Current models use "makeshift" embeddings from content features, leading to suboptimal performance. We introduced the User Sequence Imagination (USIM) framework to refine OOV embeddings by imagining user sequences and incorporating behavioral embeddings. By framing this as a reinforcement learning problem and creating a recommendation-focused reward function, USIM effectively enhances OOV recommendations. Extensive experiments demonstrate its superior performance and the ablation study further illustrates the effectiveness of USIM.

## Acknowledgment

This research was funded by the National Science Foundation of China (No.62172443) and Hunan Provincial Natural Science Foundation of China (No.2022JJ30053). This work was carried out in part using computing resources at the High-Performance Computing Center of Central South University.

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

# A  Related Works

**General Recommendation.** General recommendation systems typically predict which items users will prefer by leveraging collaborative information. Based on how collaborative information is utilized, general recommendation methods typically include approaches such as Matrix Factorization-based, Graph-based, and Sequential recommendation. Matrix Factorization-based recommendation methods derive user and item embeddings by decomposing the interaction matrix into two feature matrices using Matrix Factorization techniques [11, 23, 24]. Graph-based recommendations incorporate graph techniques [25] to model high-order relationships between users and items [4, 3, 26]. Sequential recommendation methods focus on capturing the temporal patterns in user interactions, modeling the order of users' actions to predict their future preferences [27, 28, 14, 29–32]. However, these methods are generally ineffective in addressing the OOV item,

**Out-of-Vocabulary Item Recommendation.** Out-of-vocabulary (OOV) item recommendation aims to address the problem of recommending a completely new item that has no prior user interactions with users [33, 34]. Its core idea lies in how to map the content information of the cold item into the space defined by the warm item embeddings trained from interactions with users. To achieve this goal, existing methods can be mainly categorized into two approaches. One category is the dropout model, which learns a content map function using the dropout approach [18, 35, 36, 8, 7, 9]. The other category is the generative model, which directly learns the relationship between the content of the OOV item and the IV behavior embeddings to obtain the mapping function [6, 12, 37, 5].

**Reinforcement Learning-based Recommendation**. Reinforcement learning (RL) is a famous type of learning strategy where an agent learns to make decisions by taking actions in an environment to maximize some notion of cumulative reward, which is a widely used technique for modeling human behavior [38–42]. Many studies also focus on simulating user behavior within the recommendation community. Cascading DQN [43] uses GAN to learn and simulate real users from historical interactions to obtain the reward function. In [15, 44], the simulator is trained on user historical data to simulate user feedback. In Pseudo Dyna-Q [45], a world model (user simulator) is trained by minimizing the error between online and offline rewards. Another common simulation approach is based on collaborative filtering. LIRD [46]builds a memory with (s, a, r) tuples seen in the log dataset and uses a similarity method based on cosine similarity to find the closest state-action pair to the current state and recommended action. DRR [47] and DRGR [48] use the same intuition but based on different factorizations, respectively.

# B  Details of initial model

**Loss function**    Our initial model $G$ is an MLP trained to minimize the Euclidean distance between the output of $G$ and the item embedding. Specifically, the loss function for $G$ is defined as follows:

$$\mathcal{L}(\theta) = \frac{1}{|\mathcal{I}_{iv}|} \sum_{i \in \mathcal{I}_{iv}} \|e_i - G_\theta(c_i)\|_2^2. \tag{14}$$

**Training details**    The initial model shares the same training data as USIM. We use the Adam optimizer with a learning rate of 0.001 and apply early stopping by monitoring NDCG@K on the validation set. The batch size and regularization weight are set to 1024 and 0.001, respectively.

# C  Model Details

Our complete training process is shown in Algorithm 1.

# D  Experimental Details

## D.1  Dataset Details

**Datasets**. We evaluate the USIM 's performance on cold-start items using the CiteULike and MovieLens datasets.

**Algorithm 1:** Training USIM

---

**Input:** Policy network $\pi$, Value network $\omega$, episode length $N$, initial state generator $G$, experience replay buffer $D = \emptyset$

**for** each iteration **do**

  **for** each batch **do**

    Initialize state $s_0 \leftarrow G(c_i)$

    **for** $t$ in range($N$) **do**

      Sample action $a_t$ according to Eq. equation 12

      Receive reward $r_t$ according to Eq.equation 9

      Transition to next state $s_{t+1}$ according to Eq.equation 6

      Store transition in buffer: $D \leftarrow D \bigcup (s_t, a_t, r_t, s_{t+1})$

    **end for**

    **for** each gradient step **do**

      Sample transitions from $D$ for gradient calculation

      Update the Policy network $\pi$ using the PPO loss

      Update the Value network $\omega$ with Eq. equation 13

    **end for**

  **end for**

**end for**

**Output:** Trained Policy $\pi$, Value network $\omega$

---

- **CiteULike**[2] [8] The dataset contains 5,551 users, 16,980 articles, and 204,986 interactions. On CiteULike, registered users create scientific article libraries and save articles for future reference. The goal is to leverage these libraries to recommend relevant new articles to each user. The articles are represented by 300-dimensional vectors as item content features.

- **MovieLens**[3] [17] MovieLens comprises 6,040 users, 3,883 items, and 1,000,210 interactions. The content features of items are represented using 200-dimensional vectors.

In this paper, the content features of items are represented using 200-dimensional vectors. For each dataset, 20% of items are designated as OOV items, with interactions split into a OOV validation set and testing set (1:1 ratio). Records of the remaining 80% of items are divided into training, validation, and testing sets, using an 8:1:1 ratio.

### D.2  Baseline Details

**Baselines**.To assess the effectiveness and universality of USIM, we conducted a comparative analysis with 8 leading-edge models in the domain of cold-start recommendations. This comparison was carried out across two distinct datasets. The models we benchmarked against include two main groups: (i) Dropout-based models: DropoutNet [7], MTPR [18], Heater [8], and CLCRec [9]. (ii) Generative models: DeepMusic [12], MetaEmb [19], GAR [5], and ALDI [6].

- **DeepMusic** utilizes deep neural networks to model the mean squared error (MSE) difference between generated and warm embeddings.

- **MetaEmb** trains a meta-learning-based generator for rapid convergence.

- **GAR** generates embeddings through a generative adversarial relationship with the warm recommendation model.

- **ALDI** employs distillation, using warm items as "teachers" to transfer behavioral information to cold items, referred to as "students".

- **DropoutNet** enhances cold-start robustness by randomly discarding embeddings.

- **MTPR** generates counterfactual cold embeddings considering dropout and Bayesian Personalized Ranking (BPR).

---

[2]`https://github.com/Zziwei/Heater--Cold-Start-Recommendation/tree/master/data`
[3]`https://grouplens.org/datasets/movielens/1m`

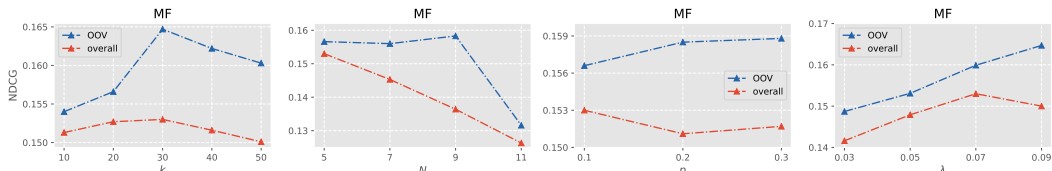

Figure 5: Hyperparameter analysis of MF backbone on CiteULike dataset

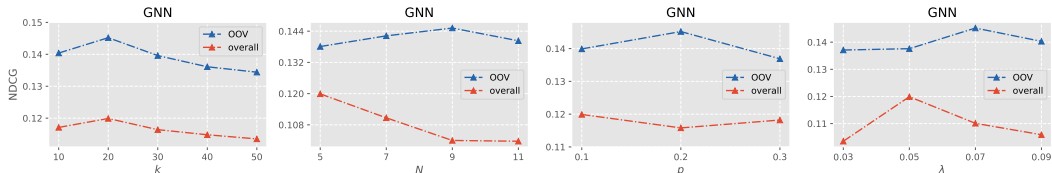

Figure 6: Hyperparameter analysis of GNN backbone on CiteULike dataset

- **Heater** improves DropoutNet by using a mix-of-experts network and considering embedding similarity.

- **CLCRec** models cold-start recommendation using contrastive learning from an information-theoretic perspective.

## E    Implementation Details

We implement the baselines using their official implementations. Specifically, for GAR, we use the updated version provided in the official repository, which is evaluated under the same CLCRec settings as used in our paper[4]. The embedding dimension is set to 200 for all models. We employ the Adam optimizer with learning rates of 0.001 for the critic and 0.0005 for the actor. Early stopping is applied by monitoring NDCG@K on the validation set. The training batch size and regularization weight are set to 1024 and 0.001, respectively.The experiment was conducted on an NVIDIA GeForce RTX 3090 with 24GB of memory. Hyperparameters are tuned using grid search, and the optimal parameters are identified for each dataset. For fairness, we use the same settings and follow the design choices in their respective articles for all baselines.

## F    Additional Experiments

We conducted a parameter analysis on the CiteULike dataset, and the results are shown in Figures 5 and 6. Except for the parameters, all other parameters exhibited the same trends on both the MF and GNN backbones.

For the parameter $k$, the performance first increased and then decreased as $k$ increased, reaching the optimal values at 30 and 20, respectively.

For the parameter $N$, the OOV recommendation performance first increased and then decreased, reaching the maximum at $N = 9$, while the overall recommendation performance consistently decreased.

For the parameter $p$, the OOV performance first increased and then decreased, while the IV performance first decreased and then increased.

For the parameter $\lambda$, on the MF backbone, the OOV performance consistently increased, while the overall performance first increased and then decreased, reaching the optimal value at $\lambda = 0.07$. On the GNN backbone, both the OOV and IV performances exhibited a trend of first increasing and then decreasing.

---

[4]https://github.com/zfnWong/GAR

# G   Details of online test

**Platform.**   We have implemented our USIM on the homepage of one of the largest e-commerce platforms, which boasts hundreds of millions of users and billions of items. The homepage features a feed recommendation system that recommends items to users. Thousands of new items are uploaded every hour.

**Framework.**   Our online implementation consists of two core components: 1. Online Recommendation; and 2. USIM Imagination. We present our framework in Figure 7 . When an OOV item is uploaded, we utilize the Large Language Model to embed the content features, including the product name and description. We then employ the USIM structure to predict the most suitable user sequence and optimize the embedding accordingly. Finally, we use the USIM-produced as the IV embeddings in the online recommendation model.

**Evaluation Metrics.**   We employed three tailored metrics to assess the performance of USIM against existing baselines:

- Page Views (OOV item PV): The number of user clicks during the OOV period.
- Page Click-Through Rate (OOV item PCTR): The ratio of clicks to impressions during the OOV period.
- Gross Merchandise Value (OOV item GMV): The total value of user purchases during the OOV period.

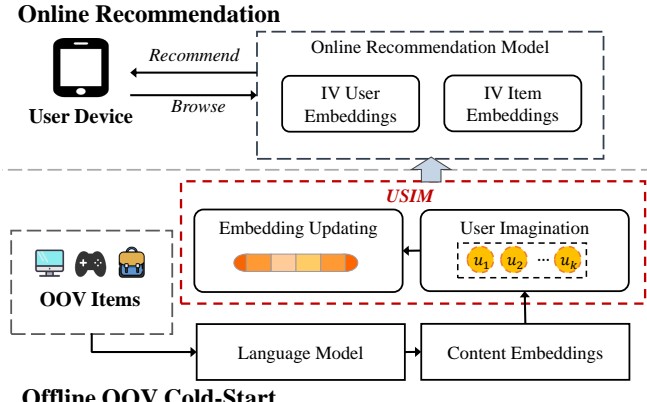

Figure 7: Overall framework of online implementation.

# H   Online Recommendation Efficiency

To evaluate the online recommendation efficiency of USIM, we recorded the LLM-based content feature extraction time (LLM Content Feature Extraction Time) and the inference time for OOV item embeddings (OOV Time) of MetaEmb, ALDI and USIM. The results are shown in Table 5.

The results of the online recommendation efficiency indicate that USIM does not serve as a bottleneck for online recommendation, as the LLM is the main time consumer while USIM's speed remains comparable to that of MetaEmb and ALDI. Additionally, the OOV recommendation is an offline, one-time process that has no impact on online recommendations, as shown in Figure 7. And given that the platform supports parallel processing, USIM computations for OOV items can be managed efficiently.

# I   Limitation

One limitation of our proposed model is the large number of hyperparameters that need to be tuned. Our model involves 4 key hyperparameters. Tuning these hyperparameters can be a time-consuming

Table 5: Results of online recommendation efficiency

| Method | LLM Content Feature Extraction Time | OOV Time |
|--------|-------------------------------------|----------|
| MetaEmb | 3.349s ± 3.214s | 0.044s ± 0.012s |
| ALDI | 3.349s ± 3.214s | 0.047s ± 0.013s |
| USIM | 3.349s ± 3.214s | 0.102s ± 0.010s |

and computationally expensive process, as it often requires extensive grid search or random search to find the optimal configuration.

Another limitation is the slow generation speed of our model due to its autoregressive generation approach. In our model, the output is generated sequentially, with each token being predicted conditioned on the previously generated tokens. This autoregressive generation process can be computationally intensive.

