# OpenReview forum: "Fine Tuning Out-of-Vocabulary Item Recommendation with User Sequence Imagination"
_NeurIPS.cc/2024/Conference — NeurIPS 2024 spotlight_

### Official Review · Reviewer_bkbE · 2024-07-09

**Soundness:** 4
**Presentation:** 3
**Contribution:** 3
**Rating:** 7
**Confidence:** 4

**Summary:**

This paper presents a novel Out-of-Vocabulary item recommendation or so-called cold-start recommendation model.

The OOV recommendation problem especially item recommendation is very important, since in short video recommendation platforms there are thousands of new post videos or new AGI videos being published every second. OOV Item recommendation are so important in dealing with the flourish of the new videos. User Sequence Imagination (USIM) provides a new paradigm for OOV recommendation to imagine behaviors of OOV items. I do like this paper, since this paper provides a totally novel and promising solution in solving OOV recommendations, of which the paradigm is quite different from traditional works. Traditional models such as generative models and dropout models do not solve the OOV recommendation fundamentally. USIM enable the OOV items to be recommended the same as In-Vocabulary items. I think this approach has the potential to fundamentally resolve the OOV issue.

**Strengths:**

This paper has the following strengths:
1. New Paradigm. The User Item Imagination is a branch new solution that can solve the OOV problem. UISM can imagine user sequences for OOV items and thus solve the OOV recommendation fundamentally.
2. Reasonable Solution. This paper proposes an RL-based USIM solution, which formulates the sequential optimization as a Markov Decision Process and introduces recommender-oriented PPO (RecPPO) to maximize the final recommendation performance of the OOV items.
3. Nice presentation and solid experiments. The Figure 1 is quite clear and insightful. Experiments are solid, comparing a wide range of baselines across various datasets.

**Weaknesses:**

1. Figure 2 can be further improved. The presentation of Figure 1 is excellent, why not keep Figure 2 in the same style?
2. There may be some typos in Equation 4. The optimization objective should be to maximize the score between e_i and e_u.
3. The fonts in Figure 4 are quite small. Besides, it’s better to modify the Table 1 to be around the main results, rather than the experimental setup.

**Questions:**

Q1. The paper mentioned that USIM has been deployed on prominent e-commerce platforms. Are there any AB-Test results or some results from the industrial side?

Q2. Could you discuss whether this model can be used for short video recommendation platforms?

**Limitations:**

Yes

---

> ### Author Rebuttal · Authors · 2024-08-07
>
> **Weakness：**
> > W1. Figure 2 can be further improved. The presentation of Figure 1 is excellent, why not keep Figure 2 in the same style?
>
> Thank you for your advice. We have revised Figure 2 in our updated PDF file. We hope this revised figure provides a clearer understanding of our work.
>
> ---
> > W2&W3. Typos and suggestions
>
> Thank you for pointing these out, we will polish our paper in the revision.
>
> ---
> > Q1: The paper mentioned that USIM has been deployed on prominent e-commerce platforms. Are there any AB-Test results or some results from the industrial side?
> Thanks for your comments, we have conducted a two-week A/B test on billion-scale recommender systems and we'd like to provide the details.
>
> **Platform** We have implemented our USIM on the homepage of one of the largest e-commerce platforms, which boasts hundreds of millions of users and billions of items. The homepage features a feed recommendation system that recommends items to users. Thousands of new items are uploaded every hour.
>
> **Framework:** Our online implementation consists of two core components: 1. Online Recommendation; and 2. USIM Imagination. We present our framework in Fig 1 in the uploaded PDF. When an OOV item is uploaded, we utilize the Large Language Model to embed the content features, including the product name and description. We then employ the USIM structure to predict the most suitable user sequence and optimize the embedding accordingly. Finally, we use the USIM-produced as the IV embeddings in the online recommendation model.
>
> **Baseline Comparisons:** We conducted an online A/B test on 5% of users for each group over two consecutive weeks. We compared USIM against three different baselines: 1. Random, 2. MetaEmb, and 3. ALDI.
> The results of the online A/B tests on the industrial platform are summarized below.
>
> | A/B Test      | OOV Item PV | OOV Item PCTR | OOV Item GMV |
> |---------------|--------------|----------------|---------------|
> | vs. Random    | +8.20%       | +2.80%         | +20.30%       |
> | vs. MetaEmb   | +6.55%       | +1.95%         | +14.95%       |
> | vs. ALDI      | +4.90%       | +1.10%         | +13.60%       |
>
> The online recommendation results demonstrate that our USIM model significantly outperforms the baselines in all three metrics. Inspired by these A/B test results, USIM is serving mainstream users and providing OOV recommendations for all newly uploaded items.
>
> ---
>
> >Q2.Could you discuss whether this model can be used for short video recommendation platforms?
>
> Thank you for your interest in applying our model in short video recommendations. Our model can be applied to OOV recommendations in short videos. We obtain the item content information, such as tags, titles, and covers, and use pre-trained models to encode them and obtain the representations. These representations are then input into the USIM model, which generates the corresponding embeddings for each OOV item.

---

> > ### Comment · Reviewer_bkbE · 2024-08-11
> > **Thanks for the rebuttal**
> >
> > Thanks for the authors' response. It addressed most of my concerns. I am happy to keep my rating for this paper.

---

> ### Author Response · Authors · 2024-08-13
> **Thank you for your reply**
>
> Thank you for your thoughtful review and feedback. We're glad our response addressed most of your concerns, and we appreciate your continued support for our paper.

---

### Official Review · Reviewer_VoqV · 2024-07-12

**Soundness:** 3
**Presentation:** 3
**Contribution:** 3
**Rating:** 7
**Confidence:** 4

**Summary:**

The authors propose the USIM (user sequence imagination) framework to tackle the out-of-vocabulary recommendation problem. The authors point out that existing OOV recommendation models such as generative and dropout models will import significant gaps between the IV items and the OOV items, since the OOV items do not have user sequence. Then USIM can imagine user sequence for OOV items to bridge the gap. Generally, I agree USIM can provide better OOV item recommendation performance.

In terms of technique details:
- The authors formulate the user sequence imagination process as a MDP, and propose a rec-PPO to optimize the imagination.
- The authors provide the State Transition Function to describe the embedding optimization process.
- The authors claim they provide RecPPO to optimize the USIM.
- The authors already implement USIM on commercial platforms to verify the effectiveness.

In summary, this paper provide a new prospective of OOV recommendation models, I recommend an accept for this paper. However, there are still some remaining issues as listed in weakness.

**Strengths:**

- The authors effectively summarize the limitations of existing models and accurately identify the core problem of the substantial gap between content features and behavioral embeddings. Their proposed User Sequence IMagination (USIM) framework is a well-reasoned solution, aiming to refine and optimize OOV item embeddings by imagining user sequences.
- By defining the user sequence imagination process as a Markov Decision Process (MDP), the authors introduce a novel perspective to the problem. The inclusion of a State Transition Function to simulate the embedding optimization of in-vocabulary (IV) items showcases a sophisticated understanding of the recommendation system dynamics and ensures a robust approach to embedding refinement.
- The authors provide extensive experiments and ablation studies to verify the effectiveness of their proposed solution. The empirical validation on benchmark datasets, along with the demonstration of superior cold-start performance and overall recommendation quality, underscores the practical applicability and robustness of the USIM framework.
- The implementation of RL-USIM on a major e-commerce platform, where it has been successfully optimizing millions of OOV items and recommending them to billions of users, highlights the real-world impact of this research. This industrial deployment not only demonstrates the scalability and effectiveness of the framework but also confirms its value in a high-stakes, large-scale environment.

**Weaknesses:**

- The RecPPO, which is a crucial component of the proposed framework, is not clearly defined in Section 3.3.2. Providing more detailed information about RecPPO would help readers better understand its role and implementation within the USIM framework.

- Figure 2, which aims to illustrate the overall USIM process, could be further refined. Enhancing this figure to more clearly and comprehensively present the USIM process would improve the paper’s clarity and help readers grasp the proposed framework's intricacies.

- Although the authors mention that USIM has been deployed on commercial platforms, there is a lack of detailed discussion on its industrial implementation and experiments. Providing insights into the deployment process, challenges faced, and specific industrial experiment results would add valuable context and strengthen the paper’s practical relevance.

**Questions:**

See weakness

**Limitations:**

Yes.

---

> ### Author Rebuttal · Authors · 2024-08-07
>
> **Weakness and Questions：**
> > W1: The RecPPO, which is a crucial component of the proposed framework, is not clearly defined in Section 3.3.2. Providing more detailed information about RecPPO would help readers better understand its role and implementation within the USIM framework.
>
> Thank you for your comments. We incorporate recommendation-specific supervision signals into PPO, referring to this enhanced approach as Recommender-Oriented PPO (RecPPO), to train our USIM. When the state representation $h^i_t$ of specific item equal to its item embedding $e_i$, expected value should be 0. So we use these supervision signals to assist in training the value network $V_{\omega}$, the specific loss function of the value network is defined as follows:
> \begin{equation}
> \begin{aligned}
>     \mathcal{L}(\omega)=\frac{1}{|B|} \sum_{(s_t, r_t, s_{t+1}) \in B} \left[ \left(r_t + \gamma V_\omega(s_{t+1}) - V_\omega(s_t)\right)^2 \right] + \frac{1}{|\mathcal{I}|} \sum_{i \in \mathcal{I}} V_\omega([{e}_i, \text{random}(0, N)])^2,
> \end{aligned}
> \end{equation}
>
> where $B$ denotes tuples sampled from the buffer pool, and $\text{random}(0, N)$ is a random number between 0 and $N$. The first term is the Temporal Difference loss used in value network training, while the second term includes our recommendation-oriented supervision signals. Regardless of previous actions, when the state representation ${h}_t^i$ matches ${e}_i$, the agent should terminate. Therefore, $\text{random}(0, N)$ serves as the countdown for each termination state.
>
>
> ---
>
> > W2: Figure 2, which aims to illustrate the overall USIM process, could be further refined. Enhancing this figure to more clearly and comprehensively present the USIM process would improve the paper’s clarity and help readers grasp the proposed framework's intricacies.
>
> Thank you for your insightful comments, we have revised and improved Fig 2, as illustrated in Fig 2 in the uploaded PDF. The updated figure provides a more detailed depiction of the USIM process, including the composition of the exploration set, the configuration of the reward function, and the state transition function. We hope this enhances your understanding of our work.
>
> ---
>
> > W3: Although the authors mention that USIM has been deployed on commercial platforms, there is a lack of detailed discussion on its industrial implementation and experiments. Providing insights into the deployment process, challenges faced, and specific industrial experiment results would add valuable context and strengthen the paper’s practical relevance.
>
> Thanks for your comments, we have conducted a two-week A/B test on billion-scale recommender systems and we'd like to provide the details.
>
> **Platform** We have implemented our USIM on the homepage of one of the largest e-commerce platforms, which boasts hundreds of millions of users and billions of items. The homepage features a feed recommendation system that recommends items to users. Thousands of new items are uploaded every hour.
>
> **Framework:** Our online implementation consists of two core components: 1. Online Recommendation; and 2. USIM Imagination. We present our framework in Fig 1 in the uploaded PDF. When an OOV item is uploaded, we utilize the Large Language Model to embed the content features, including the product name and description. We then employ the USIM structure to predict the most suitable user sequence and optimize the embedding accordingly. Finally, we use the USIM-produced as the IV embeddings in the online recommendation model.
>
> **Baseline Comparisons:** We conducted an online A/B test on 5% of users for each group over two consecutive weeks. We compared USIM against three different baselines: 1. Random, 2. MetaEmb, and 3. ALDI.
> The results of the online A/B tests on the industrial platform are summarized below.
>
> | A/B Test      | OOV Item PV | OOV Item PCTR | OOV Item GMV |
> |---------------|--------------|----------------|---------------|
> | vs. Random    | +8.20%       | +2.80%         | +20.30%       |
> | vs. MetaEmb   | +6.55%       | +1.95%         | +14.95%       |
> | vs. ALDI      | +4.90%       | +1.10%         | +13.60%       |
>
> The online recommendation results, it presents that our USIM model significantly outperforms the baselines in all three metrics. Inspired by these A/B test results, USIM is serving mainstream users and providing OOV recommendations for all newly uploaded items.

---

> > ### Comment · Reviewer_VoqV · 2024-08-13
> >
> > I will keep my score after reading the response.

---

> ### Author Response · Authors · 2024-08-13
> **Thank you for your reply**
>
> We greatly value your thorough review and kind feedback. If any additional concerns arise, please feel free to reach out to us. We’re ready to assist with more information.

---

### Official Review · Reviewer_axrp · 2024-07-16

**Soundness:** 3
**Presentation:** 3
**Contribution:** 3
**Rating:** 6
**Confidence:** 4

**Summary:**

The authors propose a novel User Sequence Imagination (USIM) fine-tuning framework. This framework can imagine the user sequences and then refine the generated OOV embeddings with user behavioral embeddings. Specifically, the authors frame the user sequence imagination as a reinforcement learning problem and develop a custom recommendation-focused reward function to evaluate the extent to which a user can help recommend the OOV items. Furthermore, they propose an embedding-driven transition function to model the embedding transition after imagining a user.

**Strengths:**

1. The author proposes a RL-based approach to optimize the embedding of out-of-vocabulary (OOV) items, allowing OOV items to benefit not only from content embedding shift but also from user interactions.

2. The author has made specific optimizations and improvements to RL methods for the recommendation scenario of out-of-vocabulary (OOV) items. These enhancements contribute to achieving better performance in recommendation systems.

**Weaknesses:**

1. The experiments conducted in this study used a limited number of datasets, and their scale was not very large. As a result, it is difficult to clearly observe the performance of USIM on large-scale datasets.

2. It is noticed that USIM does not show significant improvements over SOTA methods in certain metrics on the Movielen dataset. Additionally, the author mentions that USIM heavily relies on the selection of hyperparameters, suggesting that the performance advantage of USIM may not be significant.

3. There is a lack of theoretical analysis or experimental comparisons regarding the time efficiency of USIM. Since it is based on RL methods and processed in the form of MDP, it suggests that USIM may have lower time efficiency. Considering that it does not bring significant improvements on MovieLen, the question arises whether the high time complexity it entails is justified.

4. The author mentions that USIM has been deployed in real-world scenarios and has shown improvements. However, the experimental section of the article lacks the representation of relevant results and does not include offline evaluations on large-scale industrial datasets.

**Questions:**

1. Can the author provide the results of significance tests for the main experimental results?

2. Can the author provide a practical comparison of the experimental time between USIM and some SOTA baselines to better evaluate its efficiency?

3. The author mentions that the efficiency of USIM may not be high, while also claiming that it has been deployed in real-world scenarios. However, the deployment of recommendation systems in practical settings demands high efficiency, which contradicts the characteristics of USIM.

**Limitations:**

The authors well addressed the limitations. One limitation of USIM is the large number of hyperparameters that need to be tuned, which is time-consuming and computationally expensive. Another limitation is the slow generation speed due to its autoregressive generation paradigm.

---

> ### Author Rebuttal · Authors · 2024-08-07
>
> >W1 & W4 & Q3. Performance on large-scale datasets and online environments.
>
> Thanks for your suggestion, we conduct further experiments on other OOV recommendation datasets and on real billion-scale online recommender systems.
>
> ### Offline Experiments
> We have conducted experiments on an additional larger-scale Book-Crossing dataset, containing 92,107 users, 270,170 items, and 1,031,175 interactions. Detailed results are presented as follows:
> | Backbone | Method | Overall Recall | Overall NDCG |OOV Recall | OOV NDCG |IV Recall|IV NDCG |
> |:--:|:--:|:--:|:--:|:--:|:--:|:--:|:--:|
> |MF|MetaEmb|0.0085|0.0094|0.0137|0.0103|**0.0178**|**0.0122**|
> | |ALDI |0.0071|0.0074|0.0117|0.0066|**0.0178**|**0.0122**|
> | |DropoutNet|0.0061|0.0063|0.0102|0.0073|0.0111|0.0067|
> | |CLCRec|0.0035|0.0032|0.0131|0.0104|0.0054|0.0033|
> | |USIM|**0.0088**|**0.0097**|**0.014**|**0.011**|**0.0178**|**0.0122**|
> | |%impov.|3.53%|3.19%|2.19%|5.77%|- |- |
> |GNN|MetaEmb|0.0110|0.0109|0.0172|0.0105|**0.0305**|**0.0172**|
> | |ALDI |0.0103|0.0098|0.0125|0.0062|**0.0305**|**0.0172**|
> | |DropoutNet|0.0079|0.0087|0.0157|0.0082|0.0224|0.0133|
> | |CLCRec|0.0051|0.0046|0.0112|0.0068|0.0073|0.0044|
> | |USIM|**0.0113**|**0.0114**|**0.018**|**0.0125**|**0.0305**|**0.0172**|
> | |%impov.|2.73%|4.59%|4.44%|19.04%|- |- |
>
> Experiments on the larger-scale Book-Crossing dataset also verify the strength of USIM.
>
> ### Online Experiments
> **Platform** We have implemented our USIM on the homepage of one of the largest e-commerce platforms, which boasts hundreds of millions of users and billions of items. The homepage features a feed recommendation system that recommends items to users. Thousands of new items are uploaded every hour. The LLM service can parallelly process 20 requests of embedding, while the OOV service can parallelly process 512 items at the same time.
>
> **Framework (PDF Fig. 1):** Our online implementation consists of two core components: 1. Online Recommendation; and 2. **Offline USIM Imagination**. When an OOV item is uploaded, we utilize LLM to embed the content features and then employ the USIM structure to predict the most suitable user sequence and optimize the embedding accordingly. Finally, we use the USIM-produced embeddings as the IV embeddings for online recommendation.
>
> **Baseline Comparisons:** We conducted an online A/B test over two consecutive weeks. We compared USIM against three different baselines: 1. Random, 2. MetaEmb, and 3. ALDI.
> The results of the online A/B tests on the industrial platform are summarized below.
>
> | A/B Test      | OOV Item PV | OOV Item PCTR | OOV Item GMV |
> |:--:|:--:|:--:|:--:|
> | vs. Random    | +8.20%       | +2.80%         | +20.30%       |
> | vs. MetaEmb   | +6.55%       | +1.95%         | +14.95%       |
> | vs. ALDI      | +4.90%       | +1.10%         | +13.60%       |
>
> The online recommendation results, it presents that our USIM model significantly outperforms the baselines in all three metrics. Inspired by these A/B test results, USIM is serving mainstream users and providing OOV recommendations for all newly uploaded items.
>
> **Efficiency**
>
> | Method  | LLM Content Feature Extraction Time | OOV Time |
> |:--:|:--:|:--:|
> | MetaEmb | 3.349s $\pm$ 3.214s  | 0.044s $\pm$ 0.012   |
> | ALDI    | 3.349s $\pm$ 3.214s | 0.047s $\pm$ 0.013   |
> | USIM    | 3.349s $\pm$ 3.214s | 0.102s $\pm$ 0.010   |
>
> - **USIM is Not the Bottleneck**: The LLM is the major time consumer, while USIM is comparable to MetaEmb and ALDI.
> - **OOV Recommendation is an Offline, One-Time Process**: As shown in PDF Fig. 1, the OOV recommendation speed does not impact online recommendations, and each item undergoes the OOV process only once.
> - **OOV Recommendation Can Be Parallelized**: The online platform supports parallel processing, allowing USIM to be computed efficiently.
>
> ---
>
> >W2. Less significant improvement on MovieLens.
>
> Thanks for your comments. USIM provides a new attempt at solving the OOV recommendation problem. We admit that previous models also can achieve good performance. However, USIM can still outperform the best-performing OOV recommendation baselines, especially on CiteULike and Book-Crossing.
>
> ---
> >W3 & Q2. Efficiency analysis.
>
> Thank you for your insightful comments. We acknowledge the importance of evaluating the time efficiency of USIM, especially in comparison to SOTA baselines. Below, we present the detailed experimental time efficiency for USIM and SOTA methods.
>
> | Dataset  | Method | Total Training Time | Total Converge Epochs | Epoch Time | Inference Time |
> |:--:|:--:|:--:|:--:|:--:|:--:|
> | CiteULike | MetaEmb  | 254s  | 28  | 9.07s | 0.012s  |
> |  | ALDI | 225s  | 51   | 4.41s   | 0.013s     |
> |  | Heater | 841s | 57   | 14.75s    | 0.074s    |
> |  | CLCRec | 926s | 70   | 13.22s    | 0.070s   |
> |  | USIM | 484s | 36   | 13.44s   | 0.031s    |
> | MovieLens | MetaEmb | 415s     | 11      | 37.72s   | 0.005s  |
> |  | ALDI | 664s  | 50     | 13.28s     | 0.004s   |
> |  | Heater | 1474s  | 20    | 73.7s      | 0.072s     |
> |  | CLCRec | 3684s  | 59    | 62.44s     | 0.084s   |
> |  | USIM | 330s | 45   | 7.33s   | 0.045s   |
>
> - **USIM is Faster than Heater and CLCRec**: USIM computes embeddings only for OOV items, whereas Heater and CLCRec must compute embeddings for both OOV and IV items.
> - **USIM is Comparable with MetaEmb and ALDI**: USIM imagines sequences only for OOV items, resulting in inference times comparable to MetaEmb and ALDI.
> - **USIM is Efficient in Training**: By fundamentally addressing OOV recommendation, USIM converges in fewer epochs, making training more efficient.
>
> ---
> >Q1. Can the author provide the results of significance tests.
>
> Thanks for your suggestion. In our main table, we have already run all experiments 10 times and reported the average performance. Following your advice, we have provided the significance test results in the PDF file.

---

> > ### Comment · Reviewer_axrp · 2024-08-08
> >
> > Thank you for the detailed response, the author effectively addressed my issue, I will increase my grade.

---

> > > ### Author Response · Authors · 2024-08-08
> > > **Thanks for your reply**
> > >
> > > Thank you for your helpful review and kind support! If you have any further concerns, please feel free to reach out to us. We would be happy to provide additional details.

---

### Official Review · Reviewer_wA5P · 2024-07-30

**Soundness:** 2
**Presentation:** 2
**Contribution:** 3
**Rating:** 5
**Confidence:** 3

**Summary:**

This submission proposes a reinforcement learning framework, termed as USIM, to deal with the issue of out-of-vocabulary items in recommendation system, which is also known as cold start issue in the community. The proposed framework considers a fine-grained user sequence imagining process. Specifically, USIM formulates the imagination process of user sequence as a RL problem, and then designs recommendation-oriented reward functions for user selection. Experiments on two public datasets show that USIM achieves better performance than several baselines.

**Strengths:**

1. Cold start issue is crucial for recommendation, and a fine-grained optimization process is proposed to deal with it.

2. Code and data are available, which helps understand the implementation.

**Weaknesses:**

1. It is mentioned in the abstract that 'USIM has been deployed on a prominent e-commerce platform for months, offering recommendations for millions of OOV items and billions of users'. However, there seems no online comparison involved in the experiment. More details about online A/B test should be discussed to support the claims in the abstract.

2. There exist some statements lacking literature or empirical evidence, and some statements lacking clear description. For example:
    - In Line 47 : 'The substantial distinction between the content features and the behavioral embeddings may result in substantial discrepancies between the IV and OOV items', which is not verified with evidence.
    - Even though mentioned a lot, there is no clear definition about 'imagine user sequence', which seems not a commonly-used term.

3. The writing of the submission is too poor. The presentation of the submission is hard to follow. For example, in section 3.3.1, while the definition of MDP is described, how the MDP is related to the OOV issue is not discussed. The correspondence relation between (state, environment, action, reward) and OOV should be presented, otherwise it is quite challenging for readers to get through the proposed algorithm.

---After reading the author rebuttal---
The author rebuttal has address partial concerns. The results of A/B test have been added, which should be included in the camera ready (if accept). I have rasied my score to 5.

**Questions:**

Please refer to the weakness part.

---

> ### Author Rebuttal · Authors · 2024-08-07
>
> >W1. It is mentioned in the abstract that 'USIM has been deployed on a prominent e-commerce platform for months, offering recommendations for millions of OOV items and billions of users'. However, there seems no online comparison involved in the experiment. More details about online A/B tests should be discussed to support the claims in the abstract.
>
> Thanks for your suggestion, we have conducted a two-week A/B test and we'd like to provide the details.
>
> **Platform** We have implemented our USIM on the homepage of one of the largest e-commerce platforms, which boasts hundreds of millions of users and billions of items. The homepage features a feed recommendation system that recommends items to users. Thousands of new items are uploaded every hour.
>
> **Framework:** Our online implementation consists of two core components: 1. Online Recommendation; and 2. USIM Imagination. We present our framework in Fig 1 in the uploaded PDF. When an OOV item is uploaded, we utilize the Large Language Model to embed the content features, including the product name and description. We then employ the USIM structure to predict the most suitable user sequence and optimize the embedding accordingly. Finally, we use the USIM-produced as the IV embeddings in the online recommendation model.
>
> **Baseline Comparisons:** We conducted an online A/B test on 5% of users for each group over two consecutive weeks. We compared USIM against three different baselines: 1. Random, 2. MetaEmb, and 3. ALDI
> The results of the online A/B tests on the industrial platform are summarized below.
>
> | A/B Test      | OOV Item PV | OOV Item PCTR | OOV Item GMV |
> |---------------|--------------|----------------|---------------|
> | vs. Random    | +8.20%       | +2.80%         | +20.30%       |
> | vs. MetaEmb   | +6.55%       | +1.95%         | +14.95%       |
> | vs. ALDI      | +4.90%       | +1.10%         | +13.60%       |
>
> From the online recommendation results, it presents that our USIM model significantly outperforms the baselines in all three metrics. Further, inspired by these A/B test results, USIM is serving mainstream users and providing OOV recommendations for all newly uploaded items.
>
> ---
>
> >W2.There exist some statements lacking literature or empirical evidence, and some statements lacking clear description. For example:
> >- In Line 47 : 'The substantial distinction between the content features and the behavioral embeddings may result in substantial discrepancies between the IV and OOV items', which is not verified with evidence.
> >- Even though mentioned a lot, there is no clear definition about 'imagine user sequence', which seems not a commonly-used term.
>
>
> Thank you for your comments. We will add literature and empirical support for the gap, and provide a clearer description of the user sequence imagination. Below are our responses to the specific points raised:
>
> 1. **Evidence and Literature Support**
>
>     - **Evidence**: In recommendation systems, identical content items can elicit different user intentions. For example, two items with the same content may show vastly different engagement metrics—one may receive over a million clicks while the other gets none. This indicates that there are gaps between content features and behavioral embeddings.
>
>     - **Literature**: Papers on out-of-vocabulary (OOV) recommendations, such as GAR [1], ALDI [2], and UCC [3], demonstrate that using content features to simulate behavioral embeddings can result in differences in embedding distributions and rating distributions. These studies support the existence of gaps between content features and behavioral embeddings.
>
> 2. **Clear Definition of 'Imagine User Sequence'**
>
>     The term "imagine user sequence" refers to generating a sequence of hypothetical users using real user behavioral embeddings to refine the embeddings of an OOV item. We use "imagine" because these generated users are synthesized solely for embedding optimization. This technique allows for a more accurate representation of potential interactions, thereby improving recommendation performance for OOV items.
>
> We hope these clarifications address your concerns. We will incorporate these explanations into our revised manuscript to ensure greater clarity.
>
> [1] Chen H, Wang Z, et al. Generative adversarial framework for cold-start item recommendation. (SIGIR, 2022)
>
> [2] Huang F, Wang Z, et al. Aligning distillation for cold-start item recommendation. (SIGIR, 2023)
>
> [3] Liu T, Gao C, et al. Uncertainty-aware Consistency Learning for Cold-Start Item Recommendation. (SIGIR, 2023)
>
> ---
>
> >W3.The writing of the submission is too poor. The presentation of the submission is hard to follow. For example, in section 3.3.1, while the definition of MDP is described, how the MDP is related to the OOV issue is not discussed. The correspondence relation between (state, environment, action, reward) and OOV should be presented, otherwise it is quite challenging for readers to get through the proposed algorithm.
>
> Thanks for your comments, we will revise our manuscript to improve the overall writing quality and ensure that the explanations are clear and comprehensive. Specifically, we will:
>
> 1. Provide a detailed explanation of how MDP is related to the OOV issue.
> 2. Clearly present the correspondence between (state, environment, action, reward) and the OOV problem.
> 3. Enhance the overall readability and flow of the paper.
>
> We appreciate your constructive comments and will address these points in our revision to make the paper more accessible and easier to follow.

---

> > ### Comment · Reviewer_wA5P · 2024-08-12
> > **Rasied my score to 5 after reading the author rebuttal.**
> >
> > I have rasied my score to 5 after reading the author rebuttal.

---

> > > ### Author Response · Authors · 2024-08-13
> > > **Thank you for your reply**
> > >
> > > We sincerely appreciate your valuable review. If there are any further concerns, please don't hesitate to contact us. We're more than willing to offer additional information.

---

### Author Rebuttal · Authors · 2024-08-07

We sincerely thank all reviewers for their insightful reviews. OOV item recommendation is increasingly important in the age of information explosion and AGI. We are honored to share our findings and engage in deeper discussions with all the AC and reviewers.

We appreciate your recognition of our core strengths:

- **New OOV Item Recommendation Paradigm**: We tackle the OOV problem by imagining user sequences and optimizing with real user behavioral embeddings, rather than directly generating behavioral embeddings from content features.
- **Novel and Effective RL-Based OOV Approach**: We formulate the user sequence imagination as an MDP process, achieving promising OOV recommendation performance with an RL strategy.
- **Online Platform Application**: USIM has been implemented on one of the largest e-commerce platforms, providing OOV recommendations for all newly uploaded items on a billion-scale platform.


To better support our paper, we have added the following supplementary materials during this rebuttal process:

- **Online Implementation Framework:** We propose the introduction of our online platform and introduce the online framework that integrates seamlessly with existing systems, efficiently processing OOV items. The implemented framework is illustrated in the attached PDF.
- **A/B Test Results:**  We conducted a two-week A/B test on a large e-commerce platform to validate our claims, demonstrating significant improvements over baseline methods.
- **Improved Framework Figure:** We have enhanced the figure to clearly illustrate the components of our framework and their interactions.
- **Additional Large-Scale Dataset:** We included experiments on the Book-Crossing dataset, providing further evidence of the robustness and scalability of our approach.
- **Efficiency Experiment:** We evaluated the time efficiency of our method compared to state-of-the-art baselines, showing that our approach is not the bottleneck in processing.
- **Evidence and Literature Support:** We added references and empirical evidence to address concerns about gaps between content features and behavioral embeddings.

Thank you again for all the reviewers' suggestions, which have greatly improved our paper. For specific details, please kindly check **the corresponding responses and the attached PDF file**.

---

### Decision · Program_Chairs · 2024-09-25

**Decision:**

Accept (spotlight)

**Comment:**

Solid reviews, all reviewers recommend acceptance (even if some more enthusiastically than others). Reviewers find the problem important, the enhancements made here to be innovative / novel, and the experiments are convincing. Minor comments are made about experiments, but these seem easy enough to deal with.